# Valuing Nonuse Value of a National Forest Park with Consideration of the Local Residents' Environmental Attitudes

**Yang Yu** [1], **Erda Wang** [2,*] **and Ziang Wang** [2]

1    School of Economics and Management, Dalian Ocean University, Dalian 116023, China; yuyang770727@163.com
2    School of Economics and Management, Dalian University of Technology, Dalian 116024, China; wziang@mail.dlut.edu.cn
*    Correspondence: edwang@dlut.edu.cn; Tel.: +86-0411-8470-7090

**Abstract:** Valuing the nonuse value of a national forest park (NFP) is critically important to obtain a better understanding of its total economic value, beyond focusing solely on the recreation value. This paper estimates the nonuse value of an NFP based on the relationship between the local public's environmental attitudes and their willingness to pay (WTP). The data collected on the local residents' environmental attitudes relied on the New Ecological Paradigm (NEP). Residents' willingness to pay (WTP) for the national forest parkland protection was collected using the contingent valuation method (CVM). The nonuse value WTP was estimated using a bivariate dichotomous choice model. This model analyzed the relationship between the environmental attitude scores and WTP in order to estimate the nonuse value of the NFP of our case study site, Dalian Xijiao National Forest Park (DXNFP) in northeastern China. The results showed that DXNFP provides 20.26 CNY (3.02 USD) in nonuse value per household per year in Dalian city, which can then be translated into 140 CNY (21 USD) million annually in total.

**Keywords:** national forest park; environmental attitudes; nonuse recreational value; contingent valuation (CV); the bivariate dichotomous choice model

## 1. Introduction

Natural environmental resources such as national forest parks (NFPs) have multiple functions: safeguarding ecological balance, harmonizing mankind ecology, and boosting social economic development. Thus, NFPs carry values with multiple dimensions of environment, social, and economic [1]. However, simply recognizing the significance of those values is not adequate in policymaking and resource management; it is more critical to know the values in a quantitative manner, because only quantified value information can help policymakers implement efficient resource allocation. In recent years, researchers have started paying attention to the potential contribution NFPs can make to achieve the sustainability goals and possibly to foster inclusivity [2]. Under this context, a protected natural area such as an NFP represents an ideal place to grant accessible tourism, making the park enjoyable for everyone [3,4].

Nonuse values are the benefit or welfare gained by people who do not actually utilize the resources [5,6]. Thus, it is also called the nonuser's value [7]. Normally, the nonuse value, in economic terms, refers to the existence value and bequest value. As Krutilla [8] suggested, there are at least two reasons people might hold values unrelated to their current use of a resource. These reasons are related to preserving options for future use (existence value) and bequeathing natural resources to one's heir (bequest value). However, because the preservation of a resource for nonusers has the properties of nonexcludability and nondepletability, researchers can expect markets to fail to provide these preservation

services. There are no market transactions that reflect the nonuse values of individuals, and only direct methods of estimating nonuse values, such as the contingent valuation method, are feasible [9].

The nonuse value of an NFP has some public good characteristics, such as having no rivals and being nonexcludable [9,10]. For example, as the park produces oxygen and carbon sequestration, all the people living in the surrounding area can receive the same number of economic benefits. However, public goods are different from private goods, such as food and clothes, whose value can be easily measured using their market transaction prices. As such, empirical studies on valuing the nonuse value of public goods such as national forest parks are far less common than those valuing their use values.

Beginning in the 1970s, a number of theories and methods were developed for valuing nonuse values of environmental goods and services, including the Contingent Valuation Method (CVM) [5], Attribute-Based Method (ABM) [11], and Life Satisfaction [12]. The CVM is a well-known and widely used method for valuing nonmarket goods, especially the nonuse value [13]. The CVM has been broadly used in benefit–cost analyses of the environmental projects funded by government agencies and establishing rates for ecological compensation [14–16]. However, as the CVM is based on a hypothetical market framework, the estimated economic value has been criticized as lacking credibility. As a result, the NOAA panel provided specific recommendations regarding how a contingent valuation study should be designed and conducted to develop reliable estimates of nonuse values [17].

Economists outside the area of environmental economics and social psychologists have joined the discourse, with a particular focus on improving the credibility of the estimated economic values of the CVM. One important endeavor is establishing the noneconomic motives of nonuse values. A related undertaking is to analyze the relationship between people's attitude toward the environment and their WTP, which is then used to check the reliability of the estimated nonuse value results. Specifically, the NEP scores acquired from survey respondents are used to verify their connection with the elicited WTP. Kotchen and Reiling [18] used the NEP scores to testify the credibility of the general public's WTP for conserving the endangered species of falcons and sturgeons. The results showed that respondents upholding a positive attitude toward environmental protection would be willing to pay more money to conserve the animals than those with an unconcerned attitude. Cooper et al. [19] pointed out that it is common sense that some noneconomic motives can contribute to people's WTP for environmental protection. For example, altruism could influence people's WTP for environmental amelioration [20].

Although the nonuse value of environmental resources has long been a subject of attention for researchers in the field of economics and other disciplines, most studies focused on valuing endangered species such as birds in general [21], falcons [18], endangered sea animal species of sturgeons [18], threatened animal species [22], and dugongs [7]; independent studies on valuing nonuse values of national forest parks are less common. Bartczak [23] valued national forest resources in Poland through eliciting the general public's WTP for the policy of restricting the number of people entering national forest land areas. It was found that people who never visited national forest land areas were willing to pay 9.05 EUR for protection fees (nonuse value). Haefele et al. [24] used the CVM to estimate the total economic value (TEV) of an NFP system in the U.S.: the travel cost method (TCM) was used to determine the use-value; then, the use-value was subtracted from the TEV to derive the nonuse value of the NFP system.

In sum, few studies have evaluated the nonuse value of NFPs in a direct or unilateral manner. This phenomenon could perhaps be due to the following three reasons: First, as the nonuse value is not related to people's actual uses of NFPs, it is difficult to collect data through observing people's consumptive behaviors; second, economists prefer to ascribe the nonuse value to utility motives. However, among the studies estimating economic values of wildlife and ecosystems, it was found that over 25 percent of the respondents stated that their elicited nonuse value was not motivated by utility; rather, it originated from their social psychological ideology (e.g., all biological species have equal rights,

including human beings). This could ultimately mean that nonuse values estimated by economists are less consistent with people's real motives [18]. Third, the CVM is based on the hypothetical market condition, which usually causes an upward-biased estimated nonuse value result [25,26]. However, a recent work on developing an incentive-compatible CVM question design led Bergstrom and Randall [27] to conclude " ... that carefully designed continent valuation studies will collect a substantial body of serviceable value data, perhaps along with a minority of less reliable observations".

All policy applications, including ecological compensation [28,29], strategic decision making on NFP development [30], feasibility studies on NFP investment projects, and law enforcement regarding environmental damage compensations [31,32], require quantified economic value information on an NFP. Thus, both the recreational use-value and nonuse value are integral parts of the total economic value of an NFP. This study focuses solely on the nonuse value, as it is often more substantial than the recreational use-value. For example, according to Haefele et al.'s [24] study, the estimated total economic value (TEV) for the U.S. National Park System was 62 billion USD, of which 28.5 billion USD was the recreation use-value and 33.5 billion USD was the nonuse value. As such, the nonuse value was significantly higher than the use-value [24].

Against these theoretical and practical backdrops, this paper aims to explore the real motives of nonuse values of an NFP by analyzing the relationship between respondents' environmental attitudes and their WTP for NFP land conservation; make use of the follow-up WTP bid survey design for data collection and verify its potential advantages over those conventional CVM survey designs; and conduct an empirical measurement of nonuse values by taking the Dalian Xijiao NFP in northeastern China as an example.

The remaining content of the paper is structured as follows: Section 2 describes the research methodology including introducing the selected study area, establishing a nonuse value measurement framework, making the questionnaire deign, and conducting data collection. Section 3 presents the model results and discussion. Section 4.1 provides the study conclusions followed by Section 4.2 that addresses the study limitations.

## 2. Methodology

### 2.1. Study Area

This study was conducted in Dalian Xijiao National Forest Park (DXNFP) in northeastern China. The DXNFP is located in the southwest suburban area of Dalian city with a total land area of 5958 Ha (Figure 1). The park land is covered by dense forests and bushes and over 1 million square meters of water surface, which makes up the two reservoirs named Big West and Wang Jia Dian.

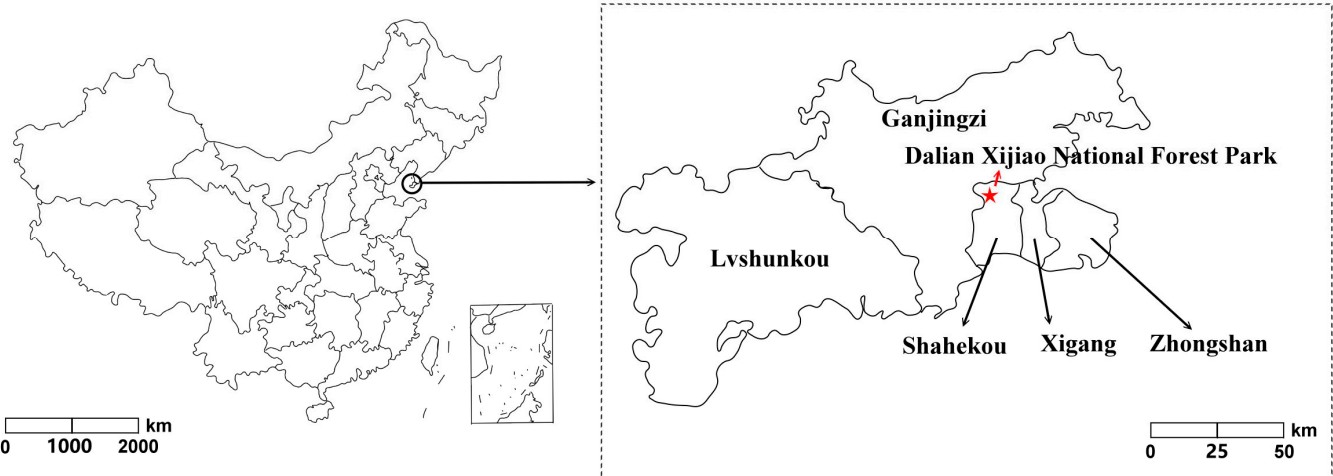

**Figure 1.** Study area of DXNFP in the northeastern China.

The DXNFP is a well-known NFP in Dalian and its surrounding areas as it offers good opportunities for people to visit nature, to breathe fresh air, and to engage in various recreational activities. For example, in 2019, the DXNFP accommodated a total of 2278 thousand visitors. Due to its free admission policy and the fast growth of the nature-based tourism demand in the local areas, in recent years, the number of visitors to DXNFP has increased tremendously, especially during holiday seasons such as the national founding days, on which the number of tourists reaches its peak level. During these days, the park management encounters the multiple pressures of traffic, congestion, and environmental damage. For example, the water quality in the West Mountain Reservoir has gradually deteriorated. Visitors often complain about the water quality, calling for speedy improvements. Nevertheless, due to the constraints of annual financial budgets, the park management does not have the money in place to take care of the water quality in the reservoirs. As such, there is an urban legend saying that DXNFP needs to sell part of the parkland to real estates or for agricultural uses to decrease the financial pressure on the park operation. Against this backdrop, this study attempts to evaluate the nonuse value of DXNFP land conservation and to explore the possibility of expanding financial resources for park operation.

Currently, the financial budgets for DXNFP completely rely on the central government, namely, the China Ministry of Finance, to provide an annual fund of roughly 42 million CNY to operate the park, which is equal to 7050 CNY (979 USD) per hectare on average. Taking 2019 as an example, the total government budget for DXNFP was 41,770 CNY (5801 USD) thousand, of which 13,560 CNY (1883 USD) thousand was used to green the park (32%), 11,100 CNY (1542 USD) thousand for infrastructure (27%), 4260 CNY (592 USD) thousand for forestry security (10%), 5010 CNY (696 USD) thousand for environmental protection (12%), and 7840 CNY (1089 USD) thousand for fire protection (19%). Obviously, the amount of money budgeted for DXNFP is far from adequate; thus, it is urgent for the park to obtain other types of money from external sources to take care of land conservation and resource protection.

*2.2. Model Design*

2.2.1. A framework for Measuring Nonuse Value of NFP

Due to the fact that the non-use value of NFPs is not associated with people's actual use of the park resources, it means that this type of value could not be estimated by observing people's consumptive behaviors as researchers usually do in valuing the recreational use-value of NFPs. As such, the contingent valuation (CV) is the only viable method that can be used in valuing non-use values [21,33–35]. Nevertheless, as the CV method has to rely on a hypothetical market condition to elicit people's maximum willingness to pay for improved environmental goods, the value estimated with the CVM is often criticized for less accuracy and the lack of reliability [17,36–39].

To deal with this problem, we resorted to the analysis of the relationship between environmental attitudes and willingness to pay. The connection between the two has been broadly studied in the area of ecological economics and environment management [40–45]. For example, Wei (2021)'s study found that the translation of environmental attitude into WTP was 61% less when the same level of environmental attitude was weakly held rather than strongly held [40]. The stronger the attitude of the environmental protection, the greater the people's willingness to pay and vice versa, whilst everything else is held constant. But, to our knowledge, no study has been conducted for the national forest park system. Thus, a Logit model was used here for this analysis. The environmental attitude measurement referred to the New Ecological Paradigm (NEP) proposed by Dunlap et al. [33], but several questions in the original NEP framework had to be adjusted in order to better align with the local people's knowledge and level of cognition on their surrounding living ecological system.

Data of the local residents' willingness to pay for DXNFP land conservation were collected using a double-bounded price bid method (see Section 3.2 for detail). As the

second price bid depends, to some degree, on the first price bid, the double-bounded price bids may exhibit some degree of correlation. In order to analyze this type of question, the general form of the bivariate dichotomous choice model must be examined. Furthermore, the bivariate Probit model was used for parameter estimation. The framework for nonuse value measurement can be described in Figure 2 below.

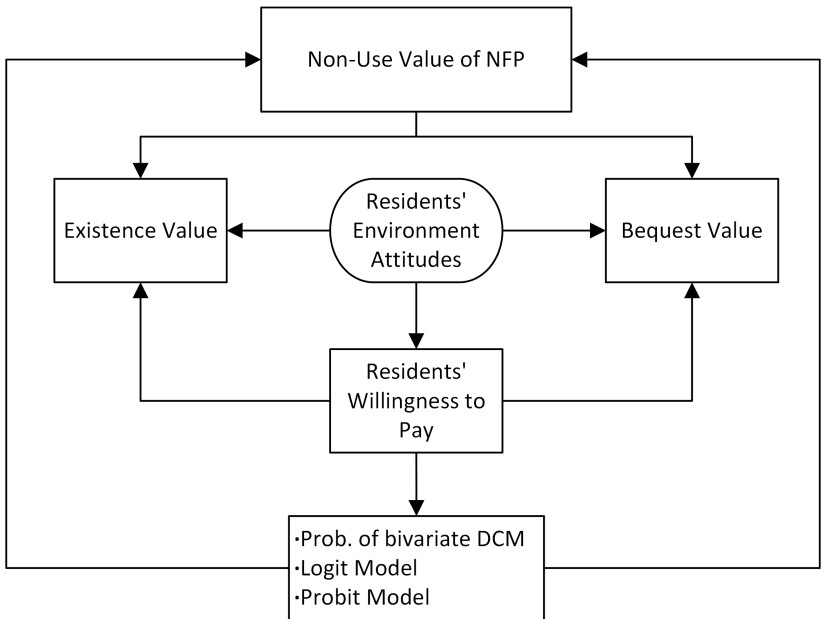

**Figure 2.** A framework for measuring nonuse value.

2.2.2. Probability Analysis of the Bivariate Dichotomous Choice Model

Let $b^1$ be the 1st bid and $b^2$ the 2nd bid; then, WTP boundaries can be expressed as follows:

(1)  For yes–no responses, $b^1 \leq WTP < b^2$;
(2)  For no–yes responses, $b^1 > WTP \geq b^2$;
(3)  For yes–yes responses, $WTP \geq b^2$;
(4)  For no–no responses, $WTP < b^2$.

Then, the most general econometric model for double-bounded data comes from Formula (1):

$$WTP_{ij} = \mu_i + e_{ij} \tag{1}$$

where $WTP_{ij}$ represents the jth respondent's willingness to pay, and i = 1, 2 represent the first and second answers, respectively. $\mu_1$ and $\mu_2$ are the means for the first and second responses, respectively, and $e_{ij}$ is the error term. We could make the same arguments using the means and depending on individual covariates: $\mu_{ij} = z_{ij}\beta$. This model incorporates the idea that, for an individual, the first and second responses to the CVM questions are different, perhaps motivated by different covariates. We assume that the mean WTP is the same for all individuals, but potentially varies across questions. To construct the likelihood function, we first derived the probability of observing each of the possible two-bid response sequences (yes–yes, yes–no, no–yes, no–no). The probability that respondent j answers yes to the first bid and no to the second is given by function (2):

$$\Pr(yes, no) = \Pr(WTP_{1j} \geq b^1, WTP_{2j} \geq b^2)$$
$$= \Pr(u_1 + e_{1j} \geq b^1, u_2 + e_{2j} \geq b^2) \tag{2}$$

The other three response sequences can be constructed analogously (see Appendices A–C).

### 2.2.3. The Bivariate Probit Model

The bivariate Probit model is a general parametric model of the double-bounded response survey. If the model is put into perspective, it can be formulated as a function of (3).

$$Y_i \ (Yes/No) = f \ (BID, GEN, AGE, EDU, INC, DET \ldots) \tag{3}$$

where $Y_i$ (1/0) is a bivariate variable (Yes/No), BID is a WTP bid, and the remaining variables are sex (GEN), age (AGE), education (EDU), family disposable income (INC), and family head (DET). According to Haab and McConnell [34], the mean WTP (WTP$_{mean}$) can be estimated by Formula (4):

$$WTP_{mean} = -\ln(1 + e^{\alpha*})/\beta \tag{4}$$

where $\alpha*$ is the combined constant and $\beta$ is the estimated coefficient of BID. Table 1 provides definitions of the variables included in the bivariate dichotomous choice model.

**Table 1.** Variable definitions of the bivariate dichotomous choice model.

| Variables | Definition |
|---|---|
| WTP | Dependent variable, respondent's estimated WTP. |
| BID | Randomly selected price bid when paying to prevent the DXNFP land loss, along with follow-up price bids. |
| GEN | Sex: 1 male, 0 female. |
| AGE | Age (unit: year) |
| EDU | Education received from school: primary school (1), middle school (2), high school (3), university (4), graduate (5). |
| INC | Family disposable income per year. |
| DET | Family head: 1 yes, 0 no. |

### 2.3. Questionnaire Design and Data Collection

#### 2.3.1. Questionnaire Design and WTP Bids

The questionnaire design included two parts: one focused on the local residents' environmental attitudes and the other the respondents' willingness to pay for DXNFP land conservation. In the environmental attitude question design, we made some adaptations for some NEP questions to attune them to the survey respondents' understanding. Thus, our environmental attitude framework (EAF) also had 15 questions, with each using Likert 5 scale scores. In comparison with the NEP, the EAF questions reflect five aspects of the respondents' environmental attitudes regarding the following: ① whether local economic growth is subject to environmental constraints; ② the existence of a center for anti-anthropology; ③ the fragility of maintaining an eco-balance; ④ the idea of socioeconomic development being subject to natural constraints; and ⑤ the possibility of ecological disaster. The WTP survey questionnaire includes three parts: part one is an introduction to the basic condition of DXNFP, part two elicits respondents' WTP for DXNFP land conservation, and part three focuses on respondents' personal traits, including sex, age, income, and education (Table 1).

Six levels of price bids were used in the survey: 5 CNY (0.74 USD), 10 CNY (1.49 USD), 15 CNY (2.24 USD), 20 CNY (2.99 USD), 30 CNY (4.48 USD), and 50 CNY (7.46 USD) (Table 2). This WTP bid combination referred to previous studies investigating WTP for restoring cultural heritage in the Shenyang municipality in northeastern China [46]. Each first bid amount was randomly assigned to an individual respondent. For example, if a respondent receives 5 CNY for their first bid, and he or she agreed to accept the price bid, then the interviewer would raise the price bid to 10 CNY for the follow-up bid, i.e., double the first price bid. If the respondent rejected the first price bid, the price bid was lowered to 2.5 CNY for the follow-up bid. As such, each respondent had two chances to express his or her WTP. This type of bidding mode doubles the chance of a price bid compared to methods with a one-time price bid only such as in the open-ended price bid mode. According to Haab and McConnell [34], double-bounded models increase the efficiency

compared to a single dichotomous choice model in three ways. First, the answer question sequences of yes–no or no–yes yield clear boundaries regarding WTP. For the no–no pairs and yes–yes pairs, there are also efficiency gains. These are because additional questions, even when they do not bound WTP completely, further constrain the distribution of the respondent's WTP. Finally, as the number of responses is increased, a given function is fitted with more observations.

**Table 2.** Design of double-bounded price bids.

| 1st Price Bid | 2nd Price Bid (Rising) | 2nd Price Bid (Falling) |
| --- | --- | --- |
| 5 CNY (0.75 USD) | 10 CNY (1.49 USD) | 2.5 CNY (0.37 USD) |
| 10 CNY (1.49 USD) | 20 CNY (2.99 USD) | 5 CNY (0.75 USD) |
| 15 CNY (2.24 USD) | 30 CNY (4.48 USD) | 7.5 CNY (1.12 USD) |
| 20 CNY (2.99 USD) | 40 CNY (5.97 USD) | 10 CNY (1.49 USD) |
| 30 CNY (4.48 USD) | 60 CNY (8.96 USD) | 15 CNY (2.24 USD) |
| 50 CNY (7.46 USD) | 100 CNY (14.93 USD) | 25 CNY (3.73 USD) |

The price-bidding process was implemented in the following manner. First, the interviewer asked the respondent how much they were willing to pay to prevent the loss of the DXNFP land area. To characterize the economic values associated with WTP, the respondent was asked to designate one of the two purposes of WTP: the existence value and the bequest value. Before answering this question, the investigators explained the definitions of each type of value. For instance, money could be paid to preserve the permanent availability of the park resources (existence value) or to safeguard the park for use by future descendants (bequest value).

### 2.3.2. Choosing Survey Respondents

The respondent selection focused on people in their 30 s or above who resided in Dalian and were the family heads. The reason for choosing this range of ages was linked to the concept of non-use value. As mentioned earlier, non-use values cover the existence value and bequest value. There is no hard rule of thumb guiding us to use to use a certain range of ages for the CVM survey. Strictly speaking, the nonuse value could be accrued by people of any adult age. Therefore, the age threshold we used in this survey was based on practicality not research evidence. According to our observations, young adults tend to pay less attention to the welfare of future generations than elderly people. This is because the young adults are less likely to have a child or children yet. Therefore, it may be too early for them to consider the non-use value of the resource.

To make the questions easier to understand, residents of the local community were invited to be involved in the questionnaire design process. The investigators took field visits to local supermarkets, city parks, and entertainment venues to ask people to read the questionnaire and provide their feedback. This process also tested whether people were able to interpret the questionnaire correctly and how long it took for a respondent to finish the survey form. On average, it took about 30 min for a person to fill out the form.

The formal survey began on National Founding Day (1 October) and lasted until the end of the month. Investigators included graduate students, who were divided into five groups with two people in each group. Five districts in Dalian city were chosen as survey areas: Zhongshan District, Xigang District, Shahekou District, Ganjingzi District, and Lvshunkou District (see Figure 1). We aimed to survey 120 households from each district; thus, a total of 600 forms were collected. After removing 35 incomplete forms, 565 forms remained. Of these, 104 were from Zhongshan, 127 from Xigang, 92 from Shahekou, 138 from Ganjingzi, and 103 from Lvshunkou. Pearson $\chi^2$ tests were made to investigate whether the sampled data from each district were unbiased against the whole district population based on the factors of age, sex, income, and education [47]. The results showed no significant difference between the sampled data and the population except for the sex ratio variable in Ganjingzi district, where the number of sampled females was higher than

that in the district population. Thus, overall, we believe that the sampled data are unbiased against the population based on selected parameters.

## 3. Results and Discussion

### 3.1. Basic Information of the Sampled Data

In the sampled data, female and male respondents account for 54% and 46%, respectively. On average, respondents are 40 years old, ranging from 30 years to 70 years old. On average, respondents receive 12 years of school education, ranging from a minimum of 5 years to a maximum of 22 years. On average, each household earns 85,000 CNY (12,690.54 USD) of disposable income annually ranging from a minimum of 18,500 CNY (2762.06 USD) to a maximum of 200,000 CNY (29,860.11 USD) with a standard deviation of 3.98. Most respondents are family heads with the authority to control the family's financial resources. The mean WTP for parkland conservation is 21 CNY (3.14 USD) with the minimum 0 to maximum 100 CNY (14.93 USD).

### 3.2. Analyzing the Local People's Environmental Attitudes

Table 3 shows the environmental attitude scores (EASs) computed from the data. Each question in Table 3 has 5 levels of ranking: 1, 2, 3, 4, 5. The higher the number, the stronger the environmental attitude. Among the even-number questions (2, 4, 6, 8, 10, 12, 14): 1 = "strongly agree" (STA); 2 = "partially agree" (SWA); 3 = "uncertain"; 4 = "partially disagree" (SWD); and 5 = "strongly disagree". By contrast, the odd-number questions (1, 3, 5, 7, 9, 11, 13) are ranked in the opposite order to the even-number questions (i.e., 1 = "strongly disagree" . . . . . . 5 = "strongly agree" STD). Thus, the minimum score is 15, the highest score is 75, and the mean score is calculated at 54.8.

**Table 3.** EAS in percentages and their correlation with the sample total scores.

| Items | STA | SWA | U | SWD | STD | $r_{i\text{-}t}$ |
|---|---|---|---|---|---|---|
| 1. The population in Dalian has approached to its land use limit. | 26.0 | 32.2 | 23.2 | 11.2 | 7.8 | 0.58 |
| 2. People in Dalian have the right to change the natural environment condition according to their own needs. | 6.7 | 25.7 | 10.8 | 32.6 | 24.3 | 0.56 |
| 3. If we interfere the sea and forest seriously, it will definitely cause disasters. | 42.2 | 38.1 | 8.5 | 8.1 | 3.0 | 0.50 |
| 4. We always have the methods to let Dalian city exist permanently. | 12.9 | 24.6 | 29.0 | 20.6 | 13.2 | 0.45 |
| 5. The economic growth in Dalian is destroying the natural environment gravely. | 41.9 | 39.4 | 6.7 | 8.9 | 3.3 | 0.59 |
| 6. If we learn how to use the environment correctly, Dalian possesses enough resources for us to use. | 31.6 | 36.0 | 15.7 | 11.2 | 5.6 | 0.40 |
| 7. Animals and people have an equal right to live in the earth. | 55.3 | 28.7 | 4.3 | 6.5 | 5.4 | 0.47 |
| 8. Conservation of nature enables us to meet the environmental chal lenges caused by economic development. | 2.1 | 10.3 | 20.9 | 32.6 | 34.2 | 0.67 |
| 9. Although science and technology have kept a fast development, we are still not able to get rid of the law of conservation of nature. | 49.8 | 41.3 | 6.3 | 1.9 | 0.9 | 0.38 |
| 10. The statement of so-called the mankind is experiencing an ecological crisis that is over exaggerated. | 7.3 | 18.1 | 25.7 | 25.5 | 23.5 | 0.71 |
| 11. The earth has unlimited resources for human to use. | 25.5 | 32.0 | 16.6 | 18.4 | 7.6 | 0.56 |
| 12. Ultimately, people are the dominator of the natural world. | 9.6 | 16.0 | 12.8 | 27.3 | 34.4 | 0.59 |
| 13. Ecological system is so fragile that its balance could be broken easily. | 38.7 | 38.5 | 9.9 | 10.5 | 2.5 | 0.59 |
| 14. In the end mankind will be able to control the natural world. | 6.2 | 18.9 | 27.5 | 26.6 | 19.9 | 0.41 |
| 15. If everything goes as it is now, Dalian will usher in a huge ecological disaster in the near future. | 23.3 | 30.1 | 27.0 | 13.4 | 6.3 | 0.67 |
| Cronbach's alpha | | | | | | 0.78 |

According to the EAS, most respondents believe that people's living activities damage the ecological balance. There is some relationship between residents' environmental attitudes and the viewpoint of opposing the 'Center of Anthropology', and they all agree that our living condition is subject to the constraints of nature conservation. However, the respondents maintain an optimistic attitude regarding people's wisdom and creativity. More than 50 percent of respondents believe there is a possibility of ecological disaster in Dalian in the near future. The last column in Table 3, labeled "$r_{i\text{-}t}$", shows the correlation coefficient of individual item score, and the total score falls in the range of 0.38–0.71, suggesting a high degree of correlation between the two. Cronbach's alpha (=0.78) verifies that the designed questions pass the internal consistency test, meaning that EAS can be used as a reliable instrument for measuring people's environmental attitudes.

### 3.3. Analyzing Relationship between Environmental Attitudes and WTP

To analyze the relationship between environmental attitudes and WTP, the Logit function (5) is used, where BID (Yes/No) is the dependent variable (Yes = 1, No = 0) and the independent variables include EAS and respondent's family income (INC). To simplify the analysis, only the first WTP bid data are used here, with a total of 340 observations.

$$\text{Yes/No} = f \text{ (BID, EAS, INC)} \tag{5}$$

Table 4 presents the model results. There is a significant positive relationship between BID and EAS and family income (INC) at ($p < 0.001$), suggesting that respondents with a strong environmental attitude are willing to accept the WTP bids. Similarly, respondents with a high family income are willing to pay more money for DXNFP land conservation.

**Table 4.** Logit model results analyzing the relationship between WTP and EAS.

| WTP | BID (Yes/No) | *p*-Value |
|---|---|---|
| Constant | −3.2906 | <0.001 |
| EAF | 0.0579 | <0.001 |
| INC | 0.1803 | <0.001 |
| Pseudo R$^2$ | 0.2157 | |
| No. Obs. | 340 | |

### 3.4. WTP Results

3.4.1. Distribution of the Double-Bounded Price Bid Responses

As the mean of WTP is computed at 21 CNY (3.14 USD) with a standard deviation of 8.25, it means that responding to WTP bids are quite scattered. According to Table 5, there are four combinations corresponding to the double-bounded dichotomous choice price bids, "yes/yes", "yes/no", "no/yes", and "no/no", and six levels of the first price bids: 5 CNY (0.75 USD), 10 CNY (1.47 USD), 15 CNY (2.24 USD), 20 CNY (2.99 USD), 30 CNY (4.48 USD), and 50 CNY (7.46 USD), respectively. The number of respondents who choose to accept the first bid and the follow-up rising price bid are 47, 102, 55, 36, 10, and 5, respectively, and the number of respondents who choose to accept the first bid and reject the follow-up price rising bid is 23, 28, 40, 30, 20, and 12, respectively. The highest frequency is 102, which occurs when accepting 10 CNY (1.47 USD) as the first price bid and 20 CNY (2.99 USD) as the follow-up increased price bid. Clearly, a trend appears: as the first price bid increases, the number of response decreases. This observation aligns with the law of demand in economics. There are 79 respondents who reject all the price bids, accounting for 14% of the data. However, this does not necessarily mean that the nonuse value of DXNFP land conservation is "0" to those respondents [48].

**Table 5.** Distribution of the double-bounded dichotomous choice WTP bids.

| 1st Bid | 2nd Bid (Rising) | 2nd Bid (Falling) | Yes/Yes | Yes/No | No/Yes | No/No | Total |
|---|---|---|---|---|---|---|---|
| 5 CNY | 10 CNY | 2.5 CNY | 47 (8.31%) | 23 (4.07%) | 25 (4.42%) | 5 (0.9%) | 100 (17.70%) |
| 10 CNY | 20 CNY | 5 CNY | 102 (18.05%) | 28 (4.96%) | 17 (3.01%) | 3 (0.5%) | 150 (26.50%) |
| 15 CNY | 30 CNY | 7.5 CNY | 55 (9.73%) | 40 (7.08%) | 10 (1.77%) | 15 (2.65%) | 120 (21.24%) |
| 20 CNY | 40 CNY | 10 CNY | 36 (6.37%) | 30 (5.31%) | 6 (1.06%) | 8 (1.42%) | 80 (14.16%) |
| 30 CNY | 60 CNY | 15 CNY | 10 (1.77%) | 20 (3.54%) | 5 (0.88%) | 25 (4.42%) | 60 (10.62%) |
| 50 CNY | 100 CNY | 25 CNY | 5 (0.88%) | 12 (2.12%) | 15 (2.65%) | 23 (4.07%) | 55 (9.73%) |
| Total | | | 255 (45.13%) | 153 (27.08%) | 78 (13.81%) | 79 (13.98) | 565 (100%) |

### 3.4.2. Results of the Bivariate Dichotomous Choice Model

The maximum likelihood estimate results of the bivariate dichotomous model are presented in Table 6. According to Table 6, BID has a negative sign and is significant ($p < 0.001$), suggesting that the probability of accepting WTP bids decreases as the price bid increases. This inverse relationship indicates the internal validity of the WTP responses. The gender variable (GEN) also reaches the same level of significance as the BID variable does at ($p < 0.001$), suggesting that a male household head is more likely to accept the price bids than a female household head. The income (INC) variable is significant ($p = 0.05$), meaning a respondent is more willing to accept the price bid for parkland protection as household income increases. Similarly, the age variable (AGE) has a significant positive effect on WTP bids at ($p = 0.045$), suggesting that, as a respondent's age rises, so does the probability of accepting the price bid. Education (EDU) has a negative sign, implying that as number of years of school education increases, the probability of accepting price bids falls. This seems to go against common wisdom. Normally, a person who receives more education tends to have a more supportive environmental attitude than the one who receives less education; thus, the former is more likely to have a high level of willingness to pay for environmental goods [21]. The negative effect of education on price bids may be attributable to the fact that income (INC) and years of education received (EDU) could have a positive relationship. This multicollinearity might cause an unexpected sign of the education variable. However, as it is not statistically significant, there is no need to pay much attention to it.

**Table 6.** Parameter estimates for the bivariate Probit model.

| Variable | B (Coeff.) | *p*-Value | Variable | B (Coeff.) | *p*-Value |
|---|---|---|---|---|---|
| $b^1$ *** | −0.168 | <0.001 | $b^2$ ** | −0.026 | <0.001 |
| INC ** | 0.085 | 0.051 | INC ** | 0.065 | 0.031 |
| EDU | −0.023 | 0.267 | EDU | −0.013 | 0.285 |
| AGE ** | 0.365 | 0.211 | AGE ** | 0.045 | 0.046 |
| GEN ** | 0.801 | 0.046 | GEN ** | 0.503 | 0.035 |
| DETN ** | 0.865 | 0.061 | DETN ** | 0.562 | 0.045 |
| Constant * | 2.577 | 0.085 | Constant * | 0.516 | 0.086 |
| ρ | 0.105 | −854.56 | | 0.11 | |
| Log-likelihood | | 463.44 | | | |
| −2ln(LR/LU) | | | | | |

Note: *** $p < 0.01$, ** $p < 0.05$, * $p < 0.1$.

In Table 6, the left three columns represent the first response; the right three columns show the second response. It appears there is a positive correlation coefficient between the two responses, but it is not significantly different from 0, suggesting no underlying

connection between the two responses. However, both price bid variables of $b^1$ and $b^2$ reach a high level of statistical significance ($p < 0.001$), suggesting an inverse relationship between price bid (BID) and the survey respondent's WTP for park land conservation. The other variables included in this model are also significant as their *p*-values are less than 0.05, except for the education (EDU) variable.

The likelihood function of the bivariate Probit shows no significant correlation between the first response and the second response, which leads us to resort to running two independent Probit models using the first-response and second-response data. When there is no correlation between the two responses, the joint estimation provides no statistical gain relative to running the two independent probit models as long as the means differ [34]. As a result, we choose to estimate two independent probit models using the first-response and second-response data, and then compute the mean of WTP using the weights calculated from the number of observations from each type of response data. Parameter estimates are presented in Tables 7 and 8.

**Table 7.** Estimated coefficients of Probit model with the first-response data (Dept. var.: WTP(y/n)).

| | B (Coeff.) | Wald | Df | *p*-Value |
|---|---|---|---|---|
| BID *** | −0.154 | 50.655 | 1.000 | <0.001 |
| INC ** | 0.078 | 0.642 | 1.000 | 0.050 |
| EDU | −0.031 | 0.711 | 1.000 | 0.267 |
| AGE ** | 0.306 | 2.645 | 1.000 | 0.045 |
| GEN *** | 0.706 | 8.567 | 1.000 | <0.001 |
| DETN ** | 0.167 | 2.412 | 1.000 | 0.032 |
| Constant | 1.577 | 1.102 | 1.000 | 0.093 |
| N | 245 | | | |
| −2Log Likelihood | 378.560 | | | |
| Cox and Snell $R^2$ | 0.365 | | | |
| Nagelkerke $R^2$ | 0.376 | | | |
| Hosmer and Lemeshow Test | $\chi^2 = 5.528$, Df = 6, $p = 0.701$ | | | |
| Overall Predictive Accuracy | 71.87 | | | |

Note: *** $p < 0.01$, ** $p < 0.05$.

**Table 8.** Estimated coefficients of probit model with the second-response data (Dept. Var.: WTP(y/n)).

| | β | Wald | Df | *p*-Value |
|---|---|---|---|---|
| BID *** | −0.524 | 47.045 | 1.000 | <0.001 |
| INC ** | 0.067 | 0.504 | 1.000 | 0.039 |
| EDU | −0.023 | 0.857 | 1.000 | 0.302 |
| AGE ** | 0.405 | 3.756 | 1.000 | 0.043 |
| GEN ** | 0.616 | 4.768 | 1.000 | 0.014 |
| DETN | 0.098 | 3.603 | 1.000 | 0.082 |
| Constant | 2.576 | 3.101 | 1.000 | 0.086 |
| N | 486 | | | |
| −2Log Likelihood | 319.560 | | | |
| Cox and Snell $R^2$ | 0.318 | | | |
| Nagelkerke $R^2$ | 0.325 | | | |
| Hosmer and Lemeshow Test | $\chi^2 = 6.514$, Df = 6, $p = 0.564$ | | | |
| Overall Predictive Accuracy | 65.42 | | | |

Note: *** $p < 0.01$, ** $p < 0.05$.

Generally, the two models generate similar results but with some differences. Results based on the first-response data are better than those based on the second-response data. Based on the first-response data, both BID and gender (GEN) variables are highly significant at *p*-value < 0.001, and the remaining variables, including income (INC), age (AGE) and

family head (DETN), are also significant at *p*-values < 0.05, but the education variable (EDU) is not significant. By contrast, based on the second-response data, all four variables of BID, INC, AGE, and GEN have significant positive effects on WTP bids but EDU and DENT.

Using Kanninen's (1993) [49] total correction classification to evaluate the goodness of fit of the bivariate probit models, the model is shown to fit the data of the first response reasonably well, with 71.87% correctness, better than the second-response data (65.42%). In addition, Hosmer and Lemeshow's test shows that the model fits the first-response data well (*p* = 0.701) and explains 37.6% of WTP variation, whereas the model using second-response data can only explain 33.8% of WTP variation.

We can use function (4) to estimate the mean WTP. According to the mean of the first-response data of 15.77 CNY (2.35 USD) and the mean of the second-response data of 30.72 CNY (4.59 USD), the weight of each response group is calculated as 0.7 and 0.3, respectively. Therefore, the nonuse value of DXNFP is 20.26 CNY (3.02 USD) per household per year, of which 60% is attributed to the existence value and 40% to the bequest value based on the survey data analysis.

There were a total of 7450.8 thousand households in Dalian by the end of 2020 according to the seventh population census. From our survey result, 7% of respondents reject any WTP bids. Thus, let us assume that the nonuse value of DXTFP for those households is 0. Certainly, this is a very conservative assumption as it is reasoned in the previous section. The 7% is tantamount to 521.6 thousand households. Deducting this from the number of households in total, there are 6929.2 thousand households remaining. If each individual household is willing to pay 20.26 CNY (3.02 USD) for DXNFP land protection annually, it can be translated into 140 million CNY (20.90 USD) of the total nonuse value.

There are few other studies from China and elsewhere with which we could compare our estimates. To provide some perspectives, Lian and Wang [46] found that the local residents of Shenyang China are willing to pay 14 CNY (2.10 USD) to restore a cultural heritage site in Shenyang Dongling National Forest Park. Sharma and Kreye [21] estimated that the residents in Pennsylvania of the U.S. are willing to pay 11.83 USD for the social values of bird conservation per household annually. Similarly, Shattuck and Poudyal et al. [47] found that, on average, an American is willing to pay 32 USD for non-consumptive activities in wildlife management areas (WMAs) in the United States. Although our estimated values differ from those derived from the United States in absolute terms, they are not off the mark considering the different circumstances, such as different currencies, divergent per capita income, and environmental ethics.

## 4. Conclusions and Study Limitations

### 4.1. Conclusions

This study evaluates nonuse values of a national forest park through examining the relationship between environmental attitudes and willingness to pay. It appears that the environmental attitude scale constitutes an internally consistent measuring instrument. Respondents who are classified with weaker, moderate, or stronger pro-environmental attitudes react differently in participating in referenda. Those with stronger pro-environmental attitudes are more likely to have a higher level of WTP for the parkland conservation than those with weaker pro-environmental attitudes. Thus, WTP estimation is sensitive to people's environmental attitudes.

The nonuse value of DXNFP is estimated as 20.26 CNY (3.02 USD) per household per year and 140 CNY (21 USD) million in total, of which 60% is attributed to the existence value and 40% to the bequest value. Obviously, the nonuse value is a significant part of the whole value of DXNFP. The findings from this research have several important implications. First, as responses to the CV survey are hypothetical, comparing them to indices of environmental attitudes provides one test of internal validity.

Second, policy makers and NFP management must take the nonuse value into consideration while NFP planning especially with regard to the investment of NFP projects and relevant resource allocation.

Third, as the nonuse value is sensible to people's environmental cognition, it is wise for governments to carry out various forms of environmental campaigns to help people better understand the significance of NFP rather than simply advertising for taking recreational activities. As people's cognition on NFP is enhanced, so is its nonuse value.

Fourth, the estimated total nonuse value of DXNFP is 3.33 times the government annual budget allocated to the park. When the use-value is considered, the economic benefits made from DXNFP operation will be much larger. This means that the park investment has achieved a very high benefit–cost ratio. However, this desired outcome should not be taken for granted and assumed to be able to last forever. If the environmental quality of the park continues to experience its current rate of deterioration owing to short financial resources and inadequate management, economic loss from park operation will be destined to take place in just a matter of time. Thus, DXNFP must look for a new financial model. Namely, the unitary governmental budget could be supplemented by a nonuse value tax levied on the local residents in order to protect the NFP for the existence value of the current generation and the bequest value to future generations.

### *4.2. Study Limitations*

The results of the joint probability distribution function of the bivariate dichotomous choice model applied in this study show that there is no statistically significant positive correlation between the first price bid response and second price bid response. Thus, we estimated the two probit models independently using the first-response and the second-response data, respectively. Nevertheless, theoretically speaking, it is possible for this relationship to be statistically significant. In this case, the estimation of independent probits on the two price bid responses would result in a loss of efficiency relative to using the bivariate probit model [34]. Therefore, it is recommended that future studies pay more attention to the relationship between the double-bounded price bid responses. If this relationship is statistically significant, the bivariate probit model must be applied to model parameter estimation.

**Author Contributions:** Conceptualization, Z.W., E.W. and Y.Y.; methodology, Z.W.; validation, Y.Y.; formal analysis, Z.W., E.W. and Y.Y.; investigation, Z.W.; resources, Y.Y. and E.W.; data collection, Z.W. and Y.Y.; writing—original draft preparation, Y.Y. and E.W.; writing—review and editing, Y.Y. and E.W.; visualization, E.W.; supervision, E.W.; project administration, E.W.; funding acquisition E.W. and Y.Y. All authors have read and agreed to the published version of the manuscript.

**Funding:** This research was funded by China's National Social Science Foundation (Ref. No. 71640035) and the Humanities and Sciences Foundation of Liaoning Province (Ref. No. JW202002).

**Data Availability Statement:** The original contributions presented in the study are included in the article. Further inquiries can be directed to the corresponding authors.

**Acknowledgments:** We are indebted to John Loomis from Colorado State University for giving us many helpful comments on preparing this paper.

**Conflicts of Interest:** The authors declare no conflict of interest.

### **Appendix A**

The *j*th contribution to the likelihood function becomes

$$L_j(\mu|b) = \Pr(\mu_1 + e_{1j} \geq b^1, u_2 + e_{2j} < b^2)^{YN}$$
$$\times \Pr(\mu_1 + e_{1j} \geq b^1, u_2 + e_{2j} \geq b^2)^{YY} \times \Pr(\mu_1 + e_{1j} < b^1, u_2 + e_{2j} < b^2)^{NN}$$
$$\times \Pr(\mu_1 + e_{1j} < b^1, u_2 + e_{2j} > b^2)^{NY}$$

where $YY = 1$ for a yes–yes answer and 0 otherwise, $NY = 1$ for a no–yes answer, etc. This formulation is referred to as the bivariate discrete choice model [31]. If the errors are assumed to be normally distributed with a mean of 0 and respective variances of $\sigma_1{}^2$ and $\sigma_2{}^2$, then $\text{WTP}_{1j}$ and $\text{WTP}_{2j}$ have a bivariate normal distribution with means $\mu_1$ and $\mu_2$,

variances of $\sigma_1^2$ and $\sigma_2^2$, and correlation coefficient $\rho$, respectively. Given the dichotomous choice responses to each question, the normally distributed model is referred to as the bivariate probit model. The likelihood function for the bivariate probit model can be derived as follows. The probability that $WTP_{1j} < b^1$ and $WTP_{2j} < b^2$, i.e., the probability of a no–no response, is

$$\Pr\left(\mu_1 + e_{1j} < b^1, u_2 + e_{2j} < b^2\right) = \varnothing_{\varepsilon_1\varepsilon_2}\left(\frac{b^1 - \mu_1}{\sigma_1}, \frac{b^2 - \mu_2}{\sigma_2}, \rho\right)$$

where $\varnothing_{\varepsilon 1 \varepsilon 2}$ is the standardized bivariate normal cumulative distribution function with zero means, unit variances, and correlation coefficient $\rho$. Similarly, the probability of no–yes, yes–no, and yes–yes responses can be derived. (For more details, please refer to Haab and McConnell [31].)

**Appendix B**

**Table A1.** Survey form of Environmental Attitude Scale (EAS). (Please choose your score.)

| Statement | Score | | | | |
|---|---|---|---|---|---|
| 1. The number of people in Dalian has reached its land use limit. | 5 | 4 | 3 | 2 | 1 |
| 2. People in Dalian have the right to change the natural environment condition according to their own needs. | 1 | 2 | 3 | 4 | 5 |
| 3. If we interfere the sea and forest seriously, it will definitely cause disasters. | 5 | 4 | 3 | 2 | 1 |
| 4. We always have the methods to let Dalian city exist permanently. | 1 | 2 | 3 | 4 | 5 |
| 5. The economic growth in Dalian is destroying the natural environment gravely. | 5 | 4 | 3 | 2 | 1 |
| 6. If we learn how to use the environment correctly, there exist enough resources for people in Dalian to use. | 1 | 2 | 3 | 4 | 5 |
| 7. Animals and people have an equal right to live in the earth. | 5 | 4 | 3 | 2 | 1 |
| 8. Conservation of nature enables us to meet the environmental challenges caused by economic development. | 1 | 2 | 3 | 4 | 5 |
| 9. Although science and technology have kept a fast speed of development, we are still not able to escape from the law of natural conservation. | 5 | 4 | 3 | 2 | 1 |
| 10. The statement of so-called 'ecological crisis' facing human kind has been greatly exaggerated. | 1 | 2 | 3 | 4 | 5 |
| 11. The earth has unlimited resources for human to use. | 5 | 4 | 3 | 2 | 1 |
| 12. Ultimately, people are the dominator of the natural world. | 1 | 2 | 3 | 4 | 5 |
| 13. Ecological system is so fragile that its balance could be broken easily. | 5 | 4 | 3 | 2 | 1 |
| 14. In the end mankind will be able to control the natural world. | 1 | 2 | 3 | 4 | 5 |
| 15. If everything goes as it is now, Dalian will usher in a huge ecological disaster in the near future. | 5 | 4 | 3 | 2 | 1 |

Each question in Table has 5 levels of ranking: 1, 2, 3, 4, 5. The higher the number, the stronger the environmental attitude. Among the even-number questions (2, 4, 6, 8, 10, 12, 14): 1 = 'strongly agree' (STA); 2 = 'partially agree' (SWA); 3 = 'uncertain'; 4 = 'partially disagree' (SWD); and 5 = 'strongly disagree'. However, the odd-number questions (1, 3, 5, 7, 9, 11, 13) are ranked in opposite order to the even-number questions (i.e., 1 = "strongly disagree" ... ... 5 = 'strongly agree' STD).

**Appendix C. Survey Form of Nonuse Values and Their Structure**

Part I. Respondent's personal information.

1. Sex: □ male, □ Femail
2. Age (Years): □ 21–30, □ 31–40, □ 41–50, □ 51–60, □ 61 and above.
3. Education received: □ Primary school, □ Middle school, □ High school, □ College or University, □ Graduate school and more.
4. Are you the family head? □ 1 yes, □ 0 No.
5. Marital status: □ Maried, □ Single, □ Cohabited.
6. Household income (Unit: CNY) per year (before tax):
   □ 10,000–20,000, □ 20,001–40,000, □ 40,001–60,000, □ 60,001–80,000, □ 80,001–100,000, □ 100,001–150,000, □ 150,001–200,000, □ 200,000–300,000, □ >300,001.

Part II. Nonuse value of the national forest park.

1.  Not only can DXNFP provide recreational use value, but also it can offer nonuse recreational value to the local people and beyond. Recreational use-value implies the well-being or happiness a person receives from participating in various recreational activities on a park site, whereas nonuse recreational value means some benefits a person can get from NFP even without having a visit to the park. Usually, nonuse value is attributed to the following three motives: option value, existence value and bequest value. Would you please tell us which motive mentioned above motivates your nonuse value?

    ① Existence value: I enjoy knowing (DXNFP) exist even if no one ever visits it. ☐ Yes or ☐ No.
    ② Bequest value: I enjoy future generations will be able to enjoy DXNFP. ☐ Yes, ☐ No.

    If the answer is "Yes" to either two questions above, continue to answer the remaining questions. If the answer is "No" to all the two questions, then stop the survey with this respondent.

2.  In recent years, financial shortage becomes a notable issue in DXNFP management due to inadequate government annual budgets allocated to the park and high financial demand from the park maintenance. In order to solve this problem, the insiders propose to sell out part of the parkland area to real estates or agricultural sectors, so that the sold land revenue could be used in supporting the park operation. Alternatively, the government could also generate the revenue through levying income tax on the local residential households. Suppose, the government decides to impose an ecological protection tax on each household in Dalian and the rationality of this tax imposition is based on people's WTP for nonuse value of the park as being referred to question 1 before. Would you be willing to pay for this ecological protection tax? ☐ Yes, ☐ No.

3.  There are six levels of the first price bid, 5 CNY, 10 CNY, 15 CNY, 20 CNY, 30 CNY, 50 CNY, with each of them associated with follow-up price bids. The tax will be paid every year in the next 10 years.

**Table A2.** Double-bounded WTP bids procedure (unit: CNY).

| 1st Bid | 2nd Bid (Rising) | 2nd Bid (Falling) | Yes/Yes | Yes/No | No/Yes | No/No |
|---|---|---|---|---|---|---|
| 5 | 10 | 2.5 | | | | |
| 10 | 20 | 5 | | | | |
| 15 | 30 | 7.5 | | | | |
| 20 | 40 | 10 | | | | |
| 30 | 60 | 15 | | | | |
| 50 | 100 | 25 | | | | |

If one accepts the 1st price bid, investigator will raise the bid with a double time, then ask the respondent's WTP again. If the first bid is rejected, investigor will lower the bid to a half of the first bid, then ask respondent's WTP again.

4.  Please indicate the weight of each type of value in percentage of your stated WTP. Existence value ☐%, Bequest value ☐%.
    If one rejects all the WTP bids, please go to answer question 5.

5.  Please give reasons why you choose to reject all the price bids?

    ① Low family income.
    ② DXNFP is owned by the government and it should pay all the money needed for park operation.
    ③ I don't trust that the government will use all the tax money for DXNFP operation.
    ④ Some other reasons, please show them: ————————————.

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
