# Peer review of "Valuing Nonuse Value of a National Forest Park with Consideration of the Local Residents’ Environmental Attitudes"

_forests, doi:10.3390/f14071487_

Round 1

Reviewer 1 Report (Previous Reviewer 3)

Thank you for considering my comments. 

I have no further objections.

Author Response

Thank you very much for satisfying with our revision work.

Reviewer 2 Report (New Reviewer)

My overall thinking is that

1. this paper is better suited to a Research Methods journal by focusing on the value of the profit model regarding the first and second price bids.

2. when removing the p=.01 significance (min p value is commonly accepted as p=.05), there is only limited significance in the model

3. the NEP scale has been modified so significantly that it it does not resemble the meaning of the NEP (a worldview) and cannot be used to infer such

4. there are considerable grammatical errors and inconsistencies.

5. other thoughts/comments are embedded within the paper.

5. 

Author Response

Reviewer 3 Report (New Reviewer)

The manuscript estimates the nonuse value of the Dalian Xijiao National Forest Park (northeastern China) based on the residents’ WTP. To this end, the Authors adopt a bivariate dichotomous choice model. The paper is interesting and well written. The methodological strategy and the discussion of the results achieved are fine. My main comments regard sections 1, 2 and 3 and are reported below.

Section1

The Authors state that “NFPs have multiple functions: safeguarding ecological balance, harmonizing mankind ecology, and boosting social economic development. Thus, NFPs provide environmental value, social economic value, etc.” Firstly, ending a sentence with “etc.” after only two points (environmental value and social economic value) is quite unusual. Instead, I suggest the Authors to refer generally to the “sustainability goals” that NFPs can contribute to, conceived in terms of environmental, social and economic dimensions.

Among the contributions to sustainability that NFPs can provide, Authors do not mention the possibility to foster inclusivity. This topic is largely debated in the literature on NFPs (see references below) and I believe that this aspect should be shortly mentioned in the manuscript.

The introduction should end with a short reminder of the paper in order to guide the reader throughout the manuscript.

Section 2

The resolution of Fig. 1 is very low. Please provide a higher resolution picture. In addition, I suggest the Authors to include an additional picture to show, at a different scale, the entire study area (through a zoom effect). 

Section 3

Subsections 3.1 and 3.2 could be merged in one single subsection. Furthermore, the Authors should explain if a pilot questionnaire was used and how they checked for the reliability of answers collected.

Minor revision:

Abstract, Line 15: “contingent” and not “continent”.

Suggested references:

“Accessibility indicator for a trails network in a Nature Park as part of the environmental assessment framework”. Environ. Impact Assess. Rev., 69, 1–15 (2018)

“Are Potential Tourists Willing to Pay More for Improved Accessibility? Preliminary Evidence from the Gargano National Park”. Land, Volume 11, Issue 1 (2022)

“Assessing the value and market attractiveness of the accessible tourism industry in Europe: A focus on major travel and leisure companies”. J. Tour. Futur., 1, 203–222 (2015)

Author Response

Reviewer 4 Report (New Reviewer)

China is a country of large numbers, so research from this market is very valuable to us. The paper contains a lot of information; however, it seems a bit messy, so the authors have to improve it.

The paper has value and makes a contribution. However, there is still room for improvement, and I give the authors few suggestions:

Fig 1: The map is quite incomprehensible. DXNFP should be located within the whole country, then within the region and Dalian city. It is necessary to draw a new map.

Line 377/382

Considering the large number of tables, I think that Tables 5.1 and 5.2 burden the text and should only be explained textually. The WTP amount in US$ should also be given in the explanation

A methodology chapter is missing, which must be introduced and explained in detail.

Line 275

It is necessary to mark the districts included in the survey on the map in order to see the spatial aspect of the research. Fig 1. should be put here.

A summary is not needed, but a conclusion. This part of the text should be reworked without repeating what was written in the previous text, but draw the necessary conclusions.

The conclusion is more general than specifically addressed. This part must be completely reworked since it has no clear conclusions and no implications.

Literature: Probably it should be useful a more literature. The literature used for this work is insufficient and uncritical.

Round 2

Reviewer 2 Report (New Reviewer)

Dear authors and editors, I had already recommended the manuscript be rejected for the reasons previously detailed (I did not recommend that it be reconsidered after a major revision). I am pleased to see that the authors have attempted to address some of these issues but the manuscript should now be submitted elsewhere for possible publication.

Author Response

Dear Reviewer,

Thank you very much for taking time reviewing our manuscript. 

Reviewer 3 Report (New Reviewer)

The manuscript looks now improved and, in my opinion, ready for publication. Of course, the final decision is up to the Editor.   

Author Response

Dear Reviewer,

We thank you so much for spending time to review our manuscript and to provide us with those very insightful and constructive comments and suggestions through out the entire manuscript review process. Your anonymous efforts and knowledge shared with us have helped us tremendously. Thank you again for accepting our revised work and kind support.

Sincerely, 

Author of the manuscript.

Reviewer 4 Report (New Reviewer)

Dear Authors,

You should continue your research and compere it with the similar researches.

Best of luck

Author Response

This manuscript is a resubmission of an earlier submission. The following is a list of the peer review reports and author responses from that submission.

Round 1

Reviewer 1 Report

The aim of the article was to estimate the non-use value of the Dalian Xijiao National Forest Park (DXNFP) in northeast China by association between the environmental attitudes of local residents and their willingness to pay (WTP)

for the Park.

The presented research topic is in line with the trend of global research on the topic non-use values ​​of environmentally valuable areas.

Application of the logit model to determine the association of ecological attitudes to

the amount of payment (WTP) for the Park and the use of a dichotomous model is correct and used in other studies on this subject in the world.

The discussion of the research results is extensive, it is sufficient to refer to e.g. to the results in the USA,

Europe, China. The specified WTP value is lower than the US results according to the authors, depends from the affluence of the respondents.

The authors of the study found that there is a significant positive correlation between local attitudes

and willingness to pay for the environment WTP.

My comment on the characteristics of the national park as a research object: in addition to the description

where it is located, there should be a map showing the location.

Summing up, the research problem undertaken in the reviewed work is very important from the scientific point of view, expanding knowledge in this field.

Reviewer 2 Report

What is the significant reason selected this site or any selection reason for this site?

Better to include the way of carrying out the field works with the local community under section 3 (data collection).

The paper could also include limitations of the study for future studies similar to this. 

Reviewer 3 Report

Estimating the nonuse value of national forest parks is a very important element in determining the total value of the environment. A necessary requirement is, of course, to establish the noneconomic motives of society in determining its value. Estimating the value of the natural environment is often made in connection with problems related to financing the functioning of various forms of nature protection.

The paper properly used the WTP method, which allows for the valuation of goods and services whose value is difficult to measure in any other way. This method is very often used to estimate nonuse values provided by the environment.

The results and conclusions obtained in the work on the basis of survey research and statistical studies are a good basis for developing research on determining the impact of various factors on the WTP value of the environment. However, according to the reviewer, the authors did not avoid several mistakes in the work, i.e.:

L. 203-2006. The respondents selected are in the age of 30 or above, but students' opinions were also used to build the survey. What was the reason if the questions were directed only to the older people?

L. 234 No explanation of "eij" in the formula.

L. 280 i tabela 5.2. The annual household disposable income is averaged ï¿¥ 95,000. Is it really? When the annual family income is from ï¿¥8,500 to ï¿¥200,000, with up to ï¿¥100,000 – 95% of respondents.

L.288-292. I don't understand, because even and uneven questions are sorted in the same way.

Table 5.5 was also not referred to.

In general, interesting and important issues related to the estimation of the society's tendency to participate in the preservation of the national park were discussed in the work. The work was written methodically correctly and carefully thought-through.
